# CREAM: Coarse-to-Fine Retrieval and Multi-modal Efficient Tuning for Document VQA

Jinxu Zhang
Harbin Institute of Technology
Harbin, China
jxzhang@ir.hit.edu.cn

Yongqi Yu
Harbin Institute of Technology
Harbin, China
yqyu@ir.hit.edu.cn

Yu Zhang*
Harbin Institute of Technology
Harbin, China
zhangyu@ir.hit.edu.cn

## Abstract

Document Visual Question Answering (DVQA) involves responding to queries based on the contents of document images. Existing works are confined to locating information within a single page and lack support for cross-page question-and-answer interactions. Furthermore, the token length limitation on model inputs can lead to the truncation of answer-relevant segments. In this study, we present CREAM, an innovative methodology that focuses on high-performance retrieval and integrates relevant multimodal document information to effectively address this critical issue. To overcome the limitations of current text embedding similarity methods, we first employ a coarse-to-fine retrieval and ranking approach. The coarse phase calculates the similarity between the query and text chunk embeddings, while the fine phase involves multiple rounds of grouping and ordering with a large language model to identify the text chunks most relevant to the query. Subsequently, integrating an attention pooling mechanism for multi-page document images into the vision encoder allows us to effectively merge the visual information of multi-page documents, enabling the multi-modal large language model (MLLM) to simultaneously process both single-page and multi-page documents. Finally, we apply various parameter-efficient tuning methods to enhance document visual question-answering performance. Experiments demonstrate that our approach secures state-of-the-art results across various document datasets.

## CCS Concepts

• **Information systems** → **Document representation**; **Question answering**; • **Applied computing** → **Document analysis**.

## Keywords

Document VQA, Retrieval Augmented Generation, Large Language Model Ranking, Multi-page Document Representation

**ACM Reference Format:**
Jinxu Zhang, Yongqi Yu, and Yu Zhang. 2024. CREAM: Coarse-to-Fine Retrieval and Multi-modal Efficient Tuning for Document VQA. In *Proceedings of the 32nd ACM International Conference on Multimedia (MM '24), October 28-November 1, 2024, Melbourne, VIC, Australia.* ACM, New York, NY, USA, 10 pages. https://doi.org/10.1145/3664647.3680750

---

*Corresponding author

## 1 Introduction

Document visual question answering holds significant practical value, enabling rapid and precise extraction of answers from extensive documents in response to user queries [29, 30, 34]. As one of the most challenging tasks in the current multimodal field, it requires understanding not only text semantics but also visual and layout information in document images. However, the majority of current methods are limited to single-page documents, demonstrating inadequate performance on multi-page documents [36, 38] as well as on lengthy content within single-page documents.

Currently, fine-tuning pre-trained visual document understanding models has yielded impressive outcomes in question-answering tasks involving visually rich documents (VRDs) [1, 15, 17, 18, 41, 42, 44]. This indicates that the integration of large-scale, unlabeled training documents during the pre-training phase of document understanding models can significantly enhance their ability to answer questions from VRDs. Despite notable advancements, these approaches heavily invest in comprehending document images. Yet, most can only process single-page documents or a fixed length of document information, leading to a constrained understanding of documents. This limitation introduces irrelevant information noise and may also result in the loss of pertinent information due to truncation. For multi-page documents, the prevalent method[36, 38] involves first identifying the document page relevant to the query and then processing it as a single-page document. This approach fundamentally lacks generalization capability, underscoring a significant journey still ahead toward practical application.

Large language models (LLMs), such as GPT-3 [5], LLaMA [37], and PaLM [6], have rapidly developed and demonstrated remarkable results across a broad spectrum of natural language processing (NLP) tasks. Recently, several methods have been explored to integrate visual features of documents into LLMs for reasoning [2, 10, 13, 43, 48]. Although certain achievements have been realized, understanding visually rich documents remains limited. The primary challenge is that relying solely on image modal fails to fully capture the semantic information of document images, and the area of multi-page document VQA remains unexplored.

In this paper, we propose CREAM, a framework that combines high-performance retrieval enhancements with multi-image, multi-modal, and efficient instruction tuning. Our approach consists of three modules: (1) an OCR engine that extracts text from document images; (2) a retrieval module that locates relevant document text chunks based on a given question; and (3) a MLLM that combines

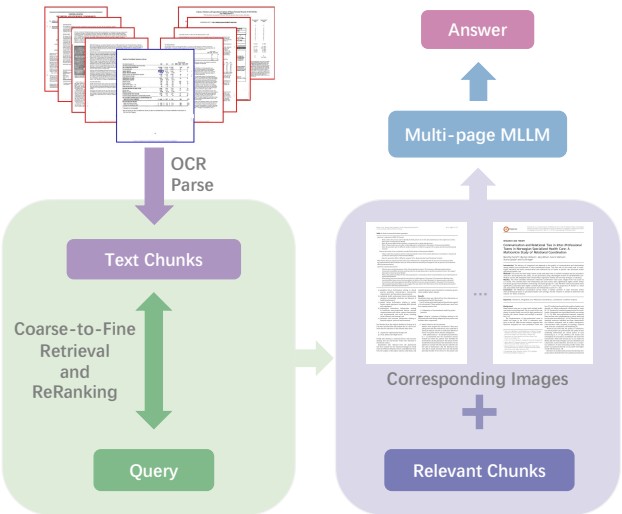

**Figure 1: The pipeline of our approach. (1) Following the input of document images and a query, the OCR tool extracts the corresponding text information and then segments the text into chunks, with each chunk corresponding to a part of the document image. (2) Utilizing our designed coarse-to-fine retrieval algorithm, we obtain text chunks pertinent to the query and their corresponding document images. (3) Incorporating the text chunks and corresponding images into our specially designed MLLM for multi-page documents, which then predicts the answer.**

multi-page document images and related document text chunks for comprehensive fine-tuning. Figure1 illustrates the specific pipeline.

Specifically, we first propose an innovative algorithm to enhance the retrieval performance. On one hand, the performance of the existing text embedding models, which is based on similarity search, has reached a bottleneck; on the other hand, the LLMs exhibit strong ranking capabilities. By integrating these approaches—initially performing coarse-grained retrieval with the text embedding model, followed by several rounds of fine-grained grouping and reranking by the LLM—the most relevant text chunks are identified. This method can not only filter out a lot of irrelevant information but also realize cross-page question answering or even multi-hop question answering, which has a strong generalization ability. Furthermore, unlike conventional text, document images encapsulate rich visual information, prompting us to utilize a vision encoder to extract this information to aid in comprehending the associated text semantics. However, current vision encoders are limited to processing single-page documents. Therefore, we have developed a vision encoder that accepts multiple document images and is integrated within an LLM. This allows for the amalgamation of relevant text chunks with corresponding document images, facilitating the concurrent consideration of text and image information pertinent to the query, thereby yielding accurate responses. Experiments were

conducted on three single-page document datasets and two multi-page document datasets, demonstrating that our method achieves state-of-the-art results in comparison with other methods.

The contributions of this paper are summarized as follows:

- We propose CREAM, a multimodal large language model designed to enhance the performance and generalization capabilities of document VQA.
- We design a coarse-to-fine retrieval algorithm to select the most relevant text chunks from document pages through embedding-based similarity retrieval, multi-round grouping, and LLM reordering, thereby enhancing the question-answering effectiveness.
- We introduce a vision encoder capable of processing multiple document images. By incorporating page information and applying attention pooling for a weighted representation of multi-page document images, we enable visual question-answering capabilities for multi-page documents through integration with a large language model.
- We achieved state-of-the-art results on two multi-page document datasets, as well as the best performance on three single-page document datasets compared to similar methods, and provided extensive ablations to each component in our method.

## 2 Related Work

### 2.1 Visually Rich Document Understanding (VRDU)

The VRDU task is aimed at interpreting content within document images, recognized as a formidable challenge. Existing approaches to VRDU can be broadly categorized by their use of Optical Character Recognition (OCR) tools. There are two primary types of models: (1) Two-Stage Models Using OCR Tools, which utilize OCR to extract text and layout information from document images. In this approach, specific pre-training tasks are designed to align visual features, layout information, and textual features within a semantic space. Examples include LayoutLMv3 [15], UDOP [35], and Docformerv2 [1], which incorporate tasks like masked image modeling and word-patch alignment, aiming to harmonize the relationship between textual content and its spatial arrangement in documents. (2) End-to-End Models Based on Image Features [17, 18]. This category's pre-training objectives typically involve text recognition tasks akin to OCR, focusing on the nuanced understanding of document images. Recently, with the advent of MLLMs [9, 23, 25, 47, 49], efforts have been made to integrate vision encoders into LLMs, thereby endowing them with image understanding capabilities. Ureader [43] has been fine-tuned on multiple document understanding datasets, including question-answering and document summarization tasks. LayoutLLM [8] enhances document understanding performance by incorporating pre-trained document vision encoders into large language models. MPLUG-DocOwl 1.5 [13] proposes unified structure learning to boost the performance of MLLMs.

While the aforementioned methods harness multimodal information from document images, they necessitate substantial resource consumption for pre-training alignment tasks. Moreover, most of

these methods can only process documents with limited information on a single page and struggle to fully grasp the nuances of document information, frequently contending with excessive noise. In our work, acknowledging that such tasks are primarily driven by textual information and augmented with image information, we leverage the reasoning capabilities of LLMs. By refining the instruction-tuning approach, we facilitate the precise generation of answers.

## 2.2 Retreival Augmented Generation (RAG)

RAG significantly enhances the input capabilities of LLMs by integrating retrieved text passages [11, 20], leading to notable improvements in knowledge-intensive tasks. This enhancement is evident post-fine-tuning and even when used with off-the-shelf LLMs [33]. Currently, RAG plays a pivotal role in addressing two key challenges associated with LLMs: the hallucination of knowledge and the need for up-to-date information. A more recent advancement [28] in the field involves instruction-tuning a Language Model (LM) by appending a fixed number of retrieved passages to the input. This approach is designed to enrich the model's context and understanding by providing additional, relevant information upfront. Furthermore, some methodologies involve jointly pre-training a retriever and an LM, which is then followed by few-shot fine-tuning on specific task datasets.

These methods, while effective in addressing open-domain questions, also encounter significant shortcomings, notably the retrieval of non-relevant information. To address this, we have innovatively applied RAG to the task of DVQA. Our approach enhances the relevance of the information retrieved by implementing a coarse-to-fine retrieval method. This method is meticulously designed to accurately isolate the specific paragraph containing the answer and to eliminate extraneous content.

## 3 Approach

We introduce CREAM, a novel approach based on LLaMA Adapter V2 [47], distinguished by its incorporation of high-performance document retrieval and visual representation of multi-page document images. Initially, we employ an OCR tool to extract text content from multi-page document images. For the text retrieval module, we introduce the coarse-to-fine retrieval method to construct relevant text chunks and corresponding document images. During the MLLM instruction-tuning phase, we expand the image encoder to capture the semantic representation of multi-page document images, integrating these into the MLLM for enhanced instruction-tuning. Ultimately, CREAM is adept at handling the task of single-page and multi-page document visual question answering.

## 3.1 Coarse-to-Fine Retrieval

RAG has recently been effectively applied across various domains, however, its application in DVQA, particularly in multi-page DVQA, remains underexplored. Given the limitation of existing models on the volume of document content they can process, prioritizing the retrieval of multi-page documents becomes essential. Furthermore, the current reliance solely on text embedding similarity for retrieval proves insufficient, as it may yield irrelevant information while

overlooking pertinent details. Consequently, we have implemented further enhancements.

Our proposed coarse-to-fine retrieval method effectively selects content and corresponding document images that are most relevant to the question. Initially, the extracted text is segmented into chunks, such as text chunks of 512 characters in length. Subsequently, the query and each text chunk are analyzed using the similarity of pre-trained text embedding to rank the text chunks. Ultimately, RankVicuna [31], a ranking pre-trained model of an LLM, groups every $m$ text chunk, and after several screening rounds, identifies the $m$ text chunks with the highest semantic match. Only the top 3 text chunks are selected as input. The algorithm's flow is depicted in Algorithm 1.

---

**Algorithm 1** Coarse-to-Fine Retrieval Algorithm

---

**Require:** *Query*: The input query
**Require:** *Chunks*: A set of document chunks, $\{C_1, C_2, \ldots, C_n\}$
**Require:** *Text_Embedding*: A pre-trained text embedding model
**Require:** *RankLLM*: A re-ranking model
**Ensure:** A re-ranked set of document chunks
  $K \leftarrow 5$ {Initialize K}
  $RankedChunks \leftarrow$ Text_Embedding($Query, Chunks$)
  $Groups \leftarrow$ Group($RankedChunks, m$) {Group every $m$ chunks}
  **while** True **do**
    $ReRankedGroups \leftarrow [\,]$
    **for** each $group \in Groups$ **do**
      $ReRankedGroup \leftarrow RankLLM(group)$
      $ReRankedGroups.append(ReRankedGroup)$
    **end for**
    $Candidates \leftarrow \bigcup_{g \in ReRankedGroups} \text{Top } K \in g$
    $ShuffledChunks \leftarrow$ Shuffle($Candidates$)
    $Groups \leftarrow$ Group($ShuffledChunks, m$)
    $K \leftarrow K - 2$
    **if** $\sum_{g \in Groups} |g| < m \vee K < 1$ **then**
      **break**
    **end if**
  **end while**
  **return** $\bigcup_{g \in Groups} g$ {Return the final set of chunks}

---

## 3.2 Multi-page Vision Encoder

Most recent multimodal LLMs employ a common framework that utilizes distinct vision and text towers to independently encode the two modalities. These encoded representations are then fused—either by projecting the image representation through one or multiple projection layers, or by direct concatenation—before being fed into LLMs. To our knowledge, all existing visual-language models only accommodate a single document image as input. To tackle accuracy and efficiency challenges, we introduce a new attention-pooling mechanism. This mechanism combines multi-page document images with page information, effectively bridging the gap between multiple image inputs and standard vision language models (VLMs). In particular, multiple images are processed by vision encoders, generating several visual representations through the vision tower

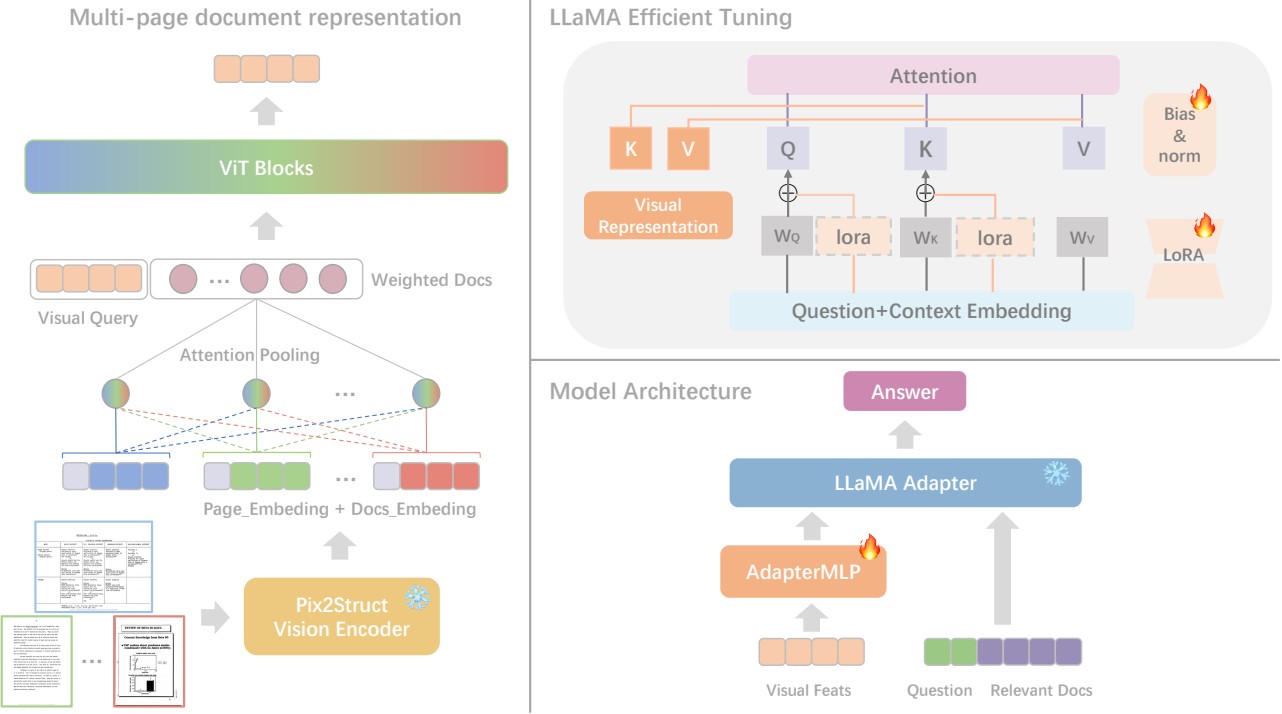

**Figure 2: Overview of our CREAM framework.CREAM features (1)a vision encoder capable of processing both single-page and multi-page document images, (2)a LLM, and (3)a projection layer linking the vision encoder to the LLM. We utilize the attention pooling mechanism to adapt the existing vision encoder, enabling it to process both single-page and multi-page documents. Upon model construction, we employ various parameter fine-tuning techniques to refine the instruction process for multi-modal document VQA.**

of any VLM, which extracts visual information from each document image. Subsequently, these visual representations, along with combined page embedding information, are fused through the attention pooling mechanism to derive the final semantic features of the multi-page document. The specific process is shown in Figure 2(left). Figure 2(right) illustrates the model architecture and parameter-efficient tuning methods we employed. The following is a detailed explanation of each module.

**Document Image Visual Representation.** For our problem setting, we introduce $I = \{D_1, D_2, \ldots, D_n\}$, comprising $N$ document images. These images are processed through a vision encoder to obtain visual representations for each document image:

$$\forall n = 1, \ldots, N : f_n = \text{Vision\_Encoder}(D_n) \tag{1}$$

where the vision encoder used from Pix2Struct [18].

Subsequently, we augment global location representations by adding them to page embeddings of corresponding document pages:

$$\forall n = 1, \ldots, N : e_n = f_n + \text{page\_embedding}_n \tag{2}$$

**Attention Pooling.** It is a weighted average of multiple document image representations.

$$E = \sum_{n=1}^{N} a_n e_n \tag{3}$$

$$a_n = \frac{\exp\{w^\top \tanh(W e_n^\top)\}}{\sum_{i=1}^{N} \exp\{w^\top \tanh(W e_i^\top)\}} \tag{4}$$

$$\sum_{n=1}^{N} a_n = 1 \tag{5}$$

Where $E$ indicates the visual vector of the document images.

**Visual Query.** Upon obtaining the weighted feature representation of multi-page document images via the attention pooling mechanism, we generate a visual query vector to capture the global feature representation of these images. Once combined with $E$ and processed through $n$ layers of vision transformer (ViT) [7] blocks, the visual query ultimately captures a deeper level of multi-page document images.

$$D_Q = Concat(Visual\_Query, E) \tag{6}$$

$$D_{global}^Q = ViTBlocks(D_Q) \tag{7}$$

$D_{global}^Q$, as the final representation of document images, is inserted into the LLM through the AdapterMLP [27] to achieve image and text alignment.

$$Visual\_Query = AdapterMLP(D_{global}^Q) \tag{8}$$

## 3.3 Multimodal Efficient Tuning

In this phase, we integrated multiple parameter efficient tuning techniques, such as LoRA [12, 14, 46], prefix tuning [19, 21, 26], and bias tuning, while incorporating the zero-initialization attention mechanism from prior methodologies [9, 40, 47], enabling us to achieve sophisticated fine-tuning effects with a minimal number of training parameters.

**LoRA.** Trainable low-rank matrices are introduced to modify the query and value matrices within the multi-head attention layer. The specific computation is implemented as follows:

Two low-rank matrices, $W_a$ and $W_b$ are initialized. These matrices have dimensions that are significantly smaller than the original query and value matrices, thereby reducing the number of trainable parameters. The original query and value matrices, denoted as $Q$ and $V$ respectively, are modified using the low-rank matrices. This modification is not a direct replacement but an additive update, which can be mathematically represented as:

$$Q' = Q + W_A Q W_B \tag{9}$$

$$V' = V + W_A V W_B \tag{10}$$

Here, $Q'$ and $V'$ represent the updated query and value matrices, respectively.

**Prefix tuning.** A prefix of length $l$ is strategically positioned preceding the key and value matrices within each multi-head attention layer. This approach effectively equates to adding $l$ additional soft prompt tokens alongside each original token for the computation of similarity measures. In this paper, we add the visual query onto the K-length adaption prompts at all $L$ inserted transformer layers. The aggregation of these calculations is conducted as follows:

$$P_l^v = P_l + Visual\_Query \tag{11}$$

$$\text{head}_i = \text{Attn}(x W_q^{(i)}, \text{concat}\left(P_k^{(i)}, P_l^v W_k^{(i)}\right),$$
$$\text{concat}\left(P_v^{(i)}, P_l^v W_v^{(i)}\right)) \tag{12}$$

where $P_l^v$ denotes the adaption prompt incorporating visual information from the given image context [47]. The length is $m$. $W_q^{(i)}, W_k^{(i)}, W_v^{(i)} \in \mathbb{R}^{d \times d_h}$, $d$ indicates embedding size, $d_h$ means attention hidden size, $P_k^{(i)}, P_v^{(i)} \in \mathbb{R}^{l \times d/N_h}$, there are $N_h$ attention heads.

To effectively manage the tasks associated with instruction-following data, same as LLaMA adapterV2 [9], we initially unfreeze all normalization layers within LLaMA. For each linear layer in the Transformer, we introduce a bias and a scale factor, both serving as learnable parameters. The input and pre-trained weights of a given linear layer are denoted as $x$ and $W$, respectively.

$$y = W \cdot x \rightarrow y = s \cdot (W \cdot x + b), \tag{13}$$

$$\text{where } b = \text{Init}(0), s = \text{Init}(1). \tag{14}$$

We initialize the bias and scale factors with zeros and ones, respectively, to stabilize the training process at the early stages.

## 4 Experiments

### 4.1 Experimental Setup

*4.1.1 Multi-page DVQA.* Our experiments are conducted on two multi-page DVQA datasets, MPDocVQA [36] and DUDE [38], both of which limited their answers to a specific page, so accurately locating the page containing the answer is crucial before attempting a response. In MPDocVQA, answers are extractive, whereas answers of DUDE encompass extractive, abstractive, list, and non-answerable types. Furthermore, the MPDocVQA dataset provides both answers and corresponding document pages, whereas DUDE only provides answers, posing challenges in locating answers within large-scale document images.

For multi-page documents, since there is no LLM to explore this task currently, we assessed two types of small pre-trained models based on their utilization of varied modal information. The first category encompasses models such as BERT [16], T5 [32], Long-Former [3], and BigBird [45], which rely on plain text information. BERT and T5 support a maximum sequence length of 512, while LongFormer and BigBird support up to 4096. The second category includes models like T5-2D [38], LayoutLMv3 [15], and HiVT5 [36], which leverage a blend of text, image, and layout information. Among these, T5-2D incorporates text and layout information with a maximum sequence length of 8192. HiVT5, designed specifically for multi-page documents, employs two methods: (1) predicting the document page likely containing the answer, similar to the single-page document process, and (2) predicting the answer for each document page, ultimately selecting the most probable one. Thus, it fundamentally addresses single-page document scenarios. This analysis reveals the limitations inherent in current pre-training models when addressing the complexities of multi-page document understanding, highlighting the need for more innovative solutions that can effectively manage the intricate dynamics of document structures across multiple pages.

Both methods utilize OCR tools for text extraction. However, existing models input the entire document content, leading to truncation when a document exceeds the maximum acceptable length. This is particularly problematic for spliced multi-page documents, as relevant information may be lost. Therefore, prioritizing the retrieval of relevant information is essential.

*4.1.2 Single-page DVQA.* We conducted experiments across three distinct types of single-page documents, briefly described as follows: the DocVQA dataset [30] consists of a significant collection of scanned and handwritten documents, with answers—usually one or more entities—being extracted from the text. Conversely, the VisualMRC dataset [34], derived from layout-rich web pages, necessitates that answers be deduced from the original text. The InfographicVQA dataset [29] showcases documents featuring various chart information and complex layouts, posing considerable challenges.

For single-page documents, since our method uses a small vision encoder and a large language model, for the sake of fairness, we compare two types of models related to our method comparison, one is small models with a pure visual or text-only modal encoder, and the other is MLLMs, which belong to the same camp as our method. In addition to the T5 [32] described above, Donut [17] and

**Table 1: Comparison results of MPDocVQA and DUDE between CREAM and the current pre-training model. In MPDocVQA, GAP indicates the gold answer page is provided, while NGAP indicates the gold answer page is not provided. Instead, the coarse-to-fine retrieval method was used to obtain relevant text and page images. Modality T, L, and V denote text, layout, and vision. Average ANLS results per question type are abbreviated as (Abs)tractive, (Ex)tractive, (N)ot-(A)nswerable, (Li)st.**

| Model | train_param | Modality | MPDocVQA | | DUDE | | | | |
| | | | NGAP | GAP | Ex | Abs | Li | NA | Over All |
|---|---|---|---|---|---|---|---|---|---|
| BERT [16] | 334M | T | 53.47 | 59.04 | 42.23 | 7.28 | 11.13 | 5.88 | 25.48 |
| T5 [32] | 223M | T | 41.80 | 68.14 | 50.49 | 47.62 | 7.56 | 63.72 | 41.80 |
| LongFormer [3] | 148M | T | 55.06 | 61.77 | 43.58 | 8.55 | 10.62 | 10.78 | 27.14 |
| BigBird [45] | 131M | T | 58.54 | 64.50 | 40.26 | 7.11 | 8.46 | 12.25 | 26.27 |
| T5-2D [36] | 770M | T+L | - | - | 55.65 | 50.81 | 5.43 | **68.62** | 46.06 |
| HiVT5 [36] | 316M | T+L+V | 62.01 | 65.72 | 17.60 | 33.94 | 6.83 | 61.76 | 23.06 |
| LayoutLMv3 [15] | 125M | T+L+V | 55.13 | 67.29 | 32.60 | 8.10 | 7.82 | 8.82 | 20.31 |
| CREAM(Ours) | 93M | T+V | **65.32** | **74.28** | **56.39** | **51.87** | **37.51** | 57.32 | **52.46** |

**Table 2: Results of comparing CREAM with existing pre-trained VDU models fine-tuned with three different categories of document visual question answering datasets. Following previous works, DocVQA and InfoVQA are evaluated by ANLS, and VisualMRC is measured by CIDEr.**

| Model | Modality | DocVQA | InfoVQA | VisualMRC |
|---|---|---|---|---|
| T5 [32] | T | 70.4 | 36.7 | 318.6 |
| Donut [17] | V | 67.5 | 11.6 | 93.9 |
| Pix2Struct [18] | V | 76.6 | 40.0 | - |
| LayoutLMv3$_{large}$ [15] | T+L+V | 83.4 | 45.1 | - |
| DocFormerv2 [1] | T+L+V | 87.7 | 48.8 | - |
| Qwen-VL [2] | V | 65.1 | 29.9 | 76.5 |
| SPHINX [24] | V | 35.8 | 24.0 | 95.3 |
| Monkey [22] | V | 66.5 | 36.1 | - |
| Ureader [43] | V | 65.4 | 42.2 | 221.7 |
| CREAM(Ours) | T+V | **79.4** | **53.6** | **377.9** |

Pix2Struct [18] focus their pre-training on text parsing from images to understand documents. Qwen-VL [2], SPHINX [10], Monkey [22], and Ureader [43] utilize off-the-shelf pre-trained vision encoders, merging them with LLMs to process multimodal information. However, document images are constituted by various data types, and existing methods fall short of comprehensively understanding them, whether through solely visual or textual analysis. Therefore, our method not only leverages the textual comprehension capabilities of large language models but also employs powerful document vision encoders for visual understanding, enhancing document visual question-answering performance by integrating both aspects without any pre-training task.

*4.1.3 Implementation Details.* In our experiments, the effectiveness of the retrieval module is important, leading us to compare several embedding models [1], including e5-large, instructor-large, and bge-large. Given the extensive volume of data involved in our testing, we ultimately selected the open-source bge-large as our retrieval model due to its robust performance. For the retrieval framework, we

utilized Langchain [2], known for its efficiency in handling complex retrieval tasks. In the group ranking phase, we set the text chunks for each group to 8.

During the instruction-tuning experiment, we opted for Pix2Struct [18] vision encoder and LLaMA2 [37] 7B as the backbone and utilized a single NVIDIA Tesla A100 80G GPU to train each dataset for 5 epochs. The batch size was set to 6, with a learning rate of 5e-5 and a weight decay of 0.02. We also set the visual prefix length to 65, applying insertions exclusively in the last 30 layers of the model. For LoRA, we set the rank r to 8. For the training phase, we chose three pages of relevant documentation as input, while for the test phase, it ranged from one to three pages.

### 4.2 Evaluation

It is noteworthy that VisualMRC is measured by CIDEr [39], and other datasets are evaluated by ANLS[4]. CIDEr is calculated as follows:

$$\text{CIDEr}_n(c_i, S_i) = \frac{1}{m} \sum_j \frac{g_n(c_i) \cdot g_n(s_{ij})}{\|g_n(c_i)\| \|g_n(s_{ij})\|} \quad (15)$$

Where $c_i$ is the candidate sentence, $S_i$ is the set of reference sentences, $m$ is the number of reference sentences, n is the length of n-gram, $g_n(c_i)$ and $g_n(s_{ij})$ are the vector of candidate sentences and reference sentences.

The calculation formula of ANLS can be expressed as:

$$\text{ANLS} = \frac{1}{N} \sum_{i=1}^{N} \left( 1 - \frac{\text{Levenshtein}(s_i, t_i)}{\max(|s_i|, |t_i|)} \right) \quad (16)$$

where N is the total number of samples. Levenshtein$(s_i, t_i)$ is the Levenshtein distance between the source string $s_i$ and the target string $t_i$ for the $i-th$ sample. In the experiment, we followed the previous approach and set the threshold to 0.5.

### 4.3 Main Results

Table 1 showcases the performance comparison of CREAM against seven models across two multi-page datasets. As demonstrated, BERT [16], LongFormer [3], BigBird [45], and LayoutLMv3 [15]

---

[1] https://huggingface.co/spaces/mteb/leaderboard

[2] https://www.langchain.com

are limited to handling extractive answer types in the MPDocVQA datasets and exhibit poor performance on other kinds of answers, such as those in the DUDE datasets. The T5-based generative models have shown improvement in abstractive answer types, yet they falter with data containing complex structures like tables and lists. Moreover, given that the DUDE dataset does not specify the page containing the answer, all existing models must accurately answer the question when the answer page is identified. Without this information, relevant content may be truncated due to input limitations, leading to suboptimal performance. For instance, HiVT5 significantly outperforms DUDE in the MPDocVQA. Although our method is influenced by certain factors, it surpasses previous approaches across three answer types. However, the LLM continues to struggle with the hallucination problem, resulting in inferior performance on non-answerable answers compared to existing methods.

Table 2 presents the performance of seven models across three single-page document datasets. CREAM exhibited superior performance in all datasets. T5 [32], Donut [17], and Pix2Struct [18] experience performance degradation as they rely solely on text modal or image modal information for document comprehension. Recently, attempts to integrate off-the-shelf image encoders into LLMs for aligning text instructions have shown that performance in document visual question answering is lacking. On the one hand, this shortfall is attributed to the superior text comprehension ability compared to the visual understanding of LLMs, on the other hand, the current image encoders exhibit limited capability in understanding document images. CREAM combines the strengths of the aforementioned models. For text comprehension, an LLM is employed, while visual understanding benefits from the superior performance of the Pix2Struct[18] vision model. This superiority, particularly against similar multimodal large language models, underscores the distinct advantages of our method.

## 4.4 Ablation Study

*4.4.1 Multi-page DVQA.* Our proposed method is more necessary for multi-page document visual question-answering. Therefore, our retrieval method is more explanatory on multi-page document datasets. As shown in Table 3, we conduct more detailed ablation experiments on the main modules in this paper.

**Effect of Text Embedding Module.** We utilize a pre-trained text embedding model to rank the text chunks of documents based on vector similarity in relation to the query. The experiment reveals that accurately identifying text chunks related to the query is challenging due to the sparse keyword semantic information in the query itself, and the possibility that each text chunk may contain relevant keywords. Consequently, distinguishing the most relevant text chunk is challenging, making reliance solely on the text embedding module insufficient.

**Effect of RankLLM Module.** Despite the large language model lacking retrieval capabilities, it exhibits strong ranking performance. This is largely attributed to its ability to assess the relevance of a query to a text chunk from a perspective akin to human language comprehension. We employed a strategy of multiple grouping rounds for repeated verification, ultimately identifying the most relevant text chunk. The experiment demonstrates that RankLLM significantly outperforms in both datasets.

**Table 3: The effect of different components in CREAM on multi-page document VQA.**

| Dataset | Vision | Text Embedding | RankLLM | ANLS |
|---------|--------|----------------|---------|------|
| MPDocVQA | ✗ | ✓ | ✗ | 58.6 |
|  | ✗ | ✓ | ✓ | 62.9 |
|  | ✓ | ✓ | ✗ | 62.8 |
|  | ✓ | ✓ | ✓ | 65.3 |
| DUDE | ✗ | ✓ | ✗ | 46.2 |
|  | ✗ | ✓ | ✓ | 49.9 |
|  | ✓ | ✓ | ✗ | 51.2 |
|  | ✓ | ✓ | ✓ | 52.5 |

**Table 4: The impact of various components in CREAM on single-page document VQA. VI represents the Vision module. RE stands for Retrieval. PB denotes Prefix and Bias tuning. LO signifies LoRA.**

|  | DocVQA | InfoVQA | VisualMRC |
|---|--------|---------|-----------|
| CREAM | 79.4 | 53.6 | 377.9 |
| w/o VI | 78.1 | 52.4 | 367.5 |
| w/o RE | 75.0 | 51.6 | 322.1 |
| w/o PB | 75.6 | 51.2 | 289.6 |
| w/o LO | 77.5 | 51.5 | 295.6 |

**Effect of Vision Module.** Our proposed multi-page document image encoder is a crucial component of this study. Given that the text within document images encompasses rich layouts and visual information, comprehending the document itself is essential. However, the capability of existing vision encoders to comprehend large-scale multi-page documents simultaneously is limited; thus, this study focuses solely on document images pertinent to the query. This encoder offers auxiliary support to the semantic information of the text. Experiments demonstrate that the vision module has a significant impact, particularly on layout-related issues.

*4.4.2 Single-page DVQA.* In this section, we conduct a detailed analysis of CREAM and its components. To evaluate the impact of individual components on model efficacy, we executed ablation studies across three categories of single-page documents, methodically omitting one module at a time, as shown in Table 4.

**Effect of Vision Module.** The ablation study on the vision module revealed that visual information is pivotal for single-page document VQA. This is attributed to the text in a document image encapsulating not only the semantic content but also its spatial position and color. Integrating this multi-modal information is essential for a deeper understanding of the document image content.

**Effect of Retrieval Module.** The majority of documents exceed 512 tokens, leading most models to employ a truncation strategy that risks omitting crucial answer segments. However, our coarse-to-fine retrieval approach accommodates the input length constraints while retaining the most pertinent sections, significantly enhancing model performance. Table 4 illustrates that the exclusion of the retrieval module results in diminished performance. Notably, During the experiment, we found through the experimental results

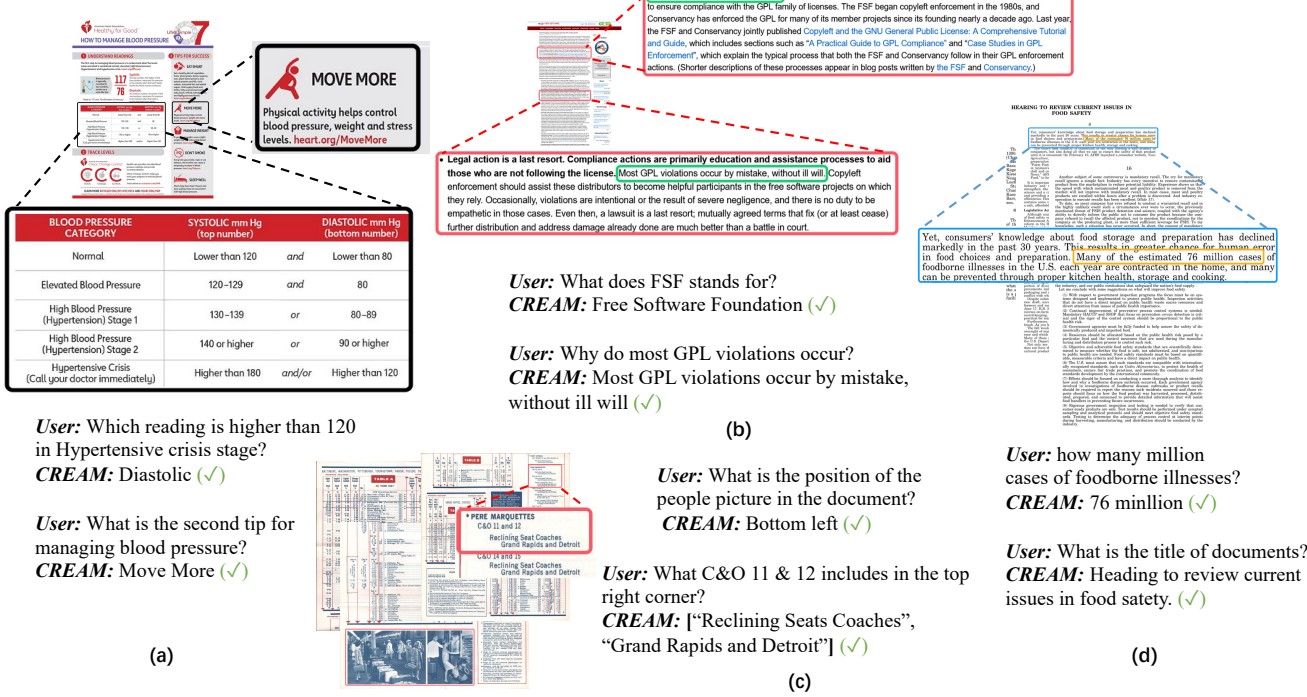

**Figure 3: Qualitative results of CREAM. Crucial regions are enlarged for clearer visualization.**

that the retrieval module exerts the most substantial impact, underscoring the efficacy of our retrieval technique across diverse test datasets.

**Effect of Prefix & Bias Tuning and LoRA.** Prefix & Bias tuning and LoRA are orthogonal approaches that do not interfere with each other. Prefix tuning allows for the injection of distinct information across various layers of the model, while Lora can further minimize the number of parameters required, without compromising on model performance. According to the experimental results, it is found that Prefix & Bias tuning and loRA have different performances on different datasets, and better results can be obtained by combining the two. It is noteworthy that visual information serves as a basis for prefix tuning. Consequently, removing the prefix module entails the removal of the vision module as well.

### 4.5 Case Study

Figure 3 shows some qualitative results produced by our CREAM on different types of documents. CREAM is adept at generating answers from documents with complex layouts (case a) and performing reasoning based on the document's content (case b). Furthermore, in multi-page documents, which typically encompass extensive textual information, CREAM can precisely pinpoint the location of an answer, as exemplified in case d. Nonetheless, the use of OCR tools in CREAM introduces certain constraints. Specifically, text extraction occurs sequentially from the top left to the bottom right, resulting in the omission of layout and visual information of the document content. Therefore, the visual information we introduce can alleviate this problem, as shown in case c. The examples

provided demonstrate that CREAM is capable of handling both single-page and multi-page documents. It not only accurately locates the relevant content of the query but also adapts its responses to the style of questions based on varying document formats.

## 5 Conclusion

This paper introduces a new model named CREAM, designed to perform single-page and multi-page DVQA simultaneously. Specifically, a coarse-to-fine retrieval algorithm is developed to extract text chunks relevant to the query. In addition, CREAM is structured as a multimodal large language model capable of processing multi-page documents and generating answers by questions, relevant text chunks, and corresponding document images. Within this framework, CREAM not only processes multimodal information but also lays a solid foundation for future multi-document image processing. Extensive experiments across two multi-page and three single-page DVQA datasets demonstrate our method's superiority over existing approaches in terms of accuracy, generalization ability, and efficiency in document visual question answering. Its limitation is that it is affected by the performance of the retrieval and ranking models, otherwise, it would achieve more desirable results.

In most document images, text layout and visual information play a key role. Therefore, for vision encoders, comprehending the multimodal information of document images is crucial. Moving forward, we aim to enhance the seamless integration and effective utilization of the model across diverse modalities.

# 6 Acknowledgements

We would like to thank the anonymous reviewers for their helpful comments. This work was supported by the National Natural Science Foundation of China (No.61976068).

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
