# OpenReview forum: "CREAM: Coarse-to-Fine Retrieval and Multi-modal Efficient Tuning for Document VQA"
_acmmm.org/ACMMM/2024/Conference — MM2024 Poster_

### Official Review · Reviewer_d4Sh · 2024-05-22

**Rating:** 5
**Confidence:** 3

**Summary:**

This paper introduces CREAM, a novel approach incorporating a coarse-to-fine retrieval strategy and a specifically designed vision encoder to address the multi-page document VQA challenges. The proposed CREAM demonstrate SOTA performance on a range of multi-page and single-page document VQA datasets.

**Strengths:**

1. The specifically designed vision encoder and coarse-to-fine retrieval strategy efficiently address the multi-page document VQA challenges.
2. CREAM outperforms numerous approaches across multi-page document VQA benchmarks such as MPDocVQA and DUDE. Furthermore, it demonstrates impressive results on single-page document VQA datasets including DocVQA, InfoVQA, and VisualMRC, while maintaining a relatively light burden with a mere 93M trainable parameters.

**Limitations:**

1. To further assess the model's adaptability, the authors may consider evaluating CREAM's performance on general VQA tasks, such as VQAv2 [1]. This would provide insights into whether the model's versatility is compromised outside its specialized domain.

2. While the comparison presented in Table 2 highlights CREAM's superiority over single-modality baselines, it is crucial to ensure a balanced evaluation, since CREAM uses both the text and visual modality. The authors may include a broader range of baseline models that also incorporate T+V information.

[1] Yash Goyal et al. Making the V in VQA Matter: Elevating the Role of Image Understanding in Visual Question Answering. CVPR, 2017.

**Suitability:**

3

---

### Official Review · Reviewer_ST53 · 2024-05-24

**Rating:** 4
**Confidence:** 3

**Summary:**

In this paper, the authors tackle the problem of document VQA, and focus especially on the problem of cross-page question-and-answer interactions.
To this end, they propose the CREAM framework that introduce :
* a coarse-to-fine retrieval method that aims to select the content and corresponding document images that are the most relevant to the question,
* a multi-page vision encoder, that is able to process both single and multi-page document images
* a multimodal efficient tuning that integrates LoRA, prefix tuning and bias tuning with zero-initialization
This framework is evaluated on Document VQA datasets with both single and multi-page documents.
Along with ablation and case studies, the authors show that CREAM outperforms previous models on multi-page and single-page documents

**Strengths:**

The paper is clear and well written.
Tackling the problem of cross-page question-and-answer interactions is an interesting research topic.
The approach is extensively described and the authors performed a lot of experiments on two kinds of datasets.

**Limitations:**

The authors could have discuss the limitations of the proposed framework.
For example, in Table 1, there is a 10 point performance's drop when the gold answer page is not known, gap that is higher for CREAM than for other approach such as HiVT5. Do the authors have an explanation for this behavior ?
Besides, the authors do not compare the performance of CREAM with LayoutLMv3 for the single-page documents, while they do it for multi-page documents. Why ? The results of the model on DocVQA are easily accessible form the paper of Huang et al. (78.73 or 83.37)
From Figure 4, we can see that the nature of the question can be different, i.e. either relying on the layout ("What is the position of the people picture in the document?") or just the textual information ("What does FSF stands for?"). It could have been interesting to have an analysis of the results according to the nature of the questions. Is there more mistakes for some particular types of questions?

Remarks :
* documents and texts are too small to be readable on Figures 2 and 4 when printed
* "large language model(MLLM)" --> "large language model (MLLM)"

**Suitability:**

3

---

### Official Review · Reviewer_vfCp · 2024-05-24

**Rating:** 4
**Confidence:** 4

**Summary:**

This paper proposed a retrieval based method for both single-page and multi-page document VQA tasks. Based on the usage of off-the-shelf Pix2Struct and Llama models, the method achieved a sota performance with only 93M parameters.

**Strengths:**

1. The paper is well written and easy to follow.
2. The proposed method is simple and efficient especially for long documents.

**Limitations:**

1. In Tab. 1, what is "the gold answer page" for MPDocVQA? Do you mean that you provide document pages with the correct page id as input? For DUDE, the results are all in NGAP or GAP? Please explain it with more details.
2. In Tab. 2, the proposed method, CREAM, takes T+V modalities input, why do you only compare with methods either in T or V? In the benchmark website (https://rrc.cvc.uab.es/?ch=17&com=evaluation&task=1), there are a lot more comparable methods with multiple modalities, e.g. LayoutLM2.0 (2020-12-22) with ANLS 86.7%. Please at least add a few multi-modal methods for a fair comparison.
3. In Tab. 3, what is the result w/ Vision, w/ Text, w/o RankLLM? Please add this important result row, and discuss it with more details.
4. The same for Tab. 4, do you have the result w/o RankLLM? The role for RankLLM is to reduce the length of text tokens while maintaining the useful information for a given question, so as to make input fit into the length limitation.
5. Please use a high contrast theme for Fig.3 for better look.

**Suitability:**

2

---

### Official Review · Reviewer_C9C1 · 2024-05-24

**Rating:** 3
**Confidence:** 3

**Summary:**

This study introduces a novel method, CREAM, aimed at enhancing the performance and generalization capabilities of Document Visual Question Answering (DVQA). The paper's multi-stage retrieval algorithm employs a coarse-to-fine similarity matching process, followed by multi-round grouping and sorting, as well as Large Language Model (LLM) reordering, effectively filtering out the text chunks most relevant to the query. This overcomes the limitations of existing methods in dealing with cross-page or multi-hop questioning. Additionally, the research designs a vision encoder capable of processing multi-page document images, integrating information from such images through an attention pooling mechanism. This enables the LLM to handle both single-page and multi-page documents concurrently, marking a significant advancement in visual encoder technology. Experimental results across multiple datasets demonstrate that this approach achieves state-of-the-art performance levels, thoroughly validating its efficacy and practical utility.

**Strengths:**

1.The article is well written, with clear logic and complete structure.

2.The experiments conducted in the paper are comprehensive and have achieved SOTA performance on multiple datasets.

**Limitations:**

1.In page 3 line 303, “Ultimately, Rank Vicuna, a ranking pre-trained model of an LLM, groups every 𝑚 text chunk.” Why are text chunks grouped during the process? Is this merely to reduce the computational load? The authors are suggested to clarify this point.

2. The authors mention employing a "coarse-to-fine retrieval method." Does this "coarse-to-fine" aspect primarily manifest in the text similarity-matching process?

3. On page 3, line 265,” To address this, we have innovatively applied RAG to the task of DVQA.” In which section do the authors apply the Retrieval-Augmented Generation (RAG) method? It is suggested that the authors dedicate a separate section in the text to outline the application of RAG explicitly.

4. The paper may not adequately address the computational costs of training the model on large-scale datasets or the efficiency in processing lengthy documents.

5. Section 5 of the experiments appears to omit comparisons with notable larger models, such as Qwen, which is also proficient in handling multi-page document processing.

**Suitability:**

2

---

### Meta-Review · Area_Chair_Tf9b · 2024-06-30

**Recommendation:** Accept (Poster)
**Confidence:** 4

**Metareview:**

The paper proposes CREAM for Document Visual Question Answering (DVQA) tasks. CREAM introduces a multi-stage retrieval algorithm using coarse-to-fine similarity matching and a vision encoder to enhance performance and generalization. The reviewers generally appreciate the clear presentation and significant experimental results of this work.

Several points requiring further clarification were noted by the reviewers. The authors addressed most of these concerns during the rebuttal stage. The overall sentiment among the reviewers leans towards acceptance. Although Reviewer C9C1 did not provide a final rating, by reviewing the authors' rebuttal myself, I reckon that most of the raised concerns have been discussed. The majority of reviewers concur that, with revisions, this paper would make a valuable contribution to the field.